# Liver Transplantation for Hepatocarcinoma: Results over Two Decades of a Transplantation Programme and Analysis of Factors Associated with Recurrence

**DOI:** 10.3390/biomedicines12061302

**Published:** 2024-06-12

**Authors:** María Martínez Burgos, Rocío González Grande, Susana López Ortega, Inmaculada Santaella Leiva, Jesús de la Cruz Lombardo, Julio Santoyo Santoyo, Miguel Jiménez Pérez

**Affiliations:** 1Liver Transplant Unit, Digestive System Department, Hospital Regional Universitario de Málaga, 29010 Malaga, Spain; rociogongrande@hotmail.com (R.G.G.); susanalopezortega@yahoo.es (S.L.O.); inmasantaella@gmail.com (I.S.L.); jesuspilarmail@telefonica.net (J.d.l.C.L.); miguel.jimenez.sspa@juntadeandalucia.es (M.J.P.); 2Instituto de Investigación Biomedica de Plataforma en Nanomedicina—IBIMA Plataforma Bionand, 29590 Malaga, Spain; julio.santoyo.sspa@juntadeandalucia.es; 3Liver Transplant Unit, General Surgery and Digestive System Department, Hospital Regional Universitario de Málaga, 29010 Malaga, Spain

**Keywords:** liver transplantation, hepatocarcinoma, recurrence

## Abstract

Background: In recent years, many studies have attempted to develop models to predict the recurrence of hepatocarcinoma after liver transplantation. Method: A single-centre, retrospective cohort study analysed patients receiving transplants due to hepatocarcinoma during the 20 years of the transplant programme. We analysed patient survival, hepatocarcinoma recurrence and the influence of the different factors described in the literature as related to hepatocarcinoma recurrence. We compared the results of previous items between the first and second decades of the transplantation programme (1995–2010 and 2010–2020). Results: Of 265 patients, the patient survival rate was 68% at 5 years, 58% at 10 years, 45% at 15 years and 34% at 20 years. The overall recurrence rate of hepatocarcinoma was 14.5%, without differences between periods. Of these, 54% of recurrences occurred early, in the first two years after transplantation. Of the parameters analysed, an alpha-fetoprotein level of >16 ng/mL, the type of immunosuppression used and the characteristics of the pathological anatomy of the explant were significant. A trend towards statistical significance was identified for the number of nodules and the size of the largest nodule. Logistic regression analysis was used to develop a model with a sensitivity of 85.7% and a specificity of 35.7% to predict recurrences in our cohort. Regarding the comparison between periods, the survival and recurrence rates of hepatocarcinoma were similar. The impact of the factors analysed in both decades was similar. Conclusions: Most recurrences occur during the first two years post-transplantation, so closer follow-ups should be performed during this period, especially in those patients where the model predicts a high risk of recurrence. The detection of patients at higher risk of recurrence allows for closer follow-up and may, in the future, make them candidates for adjuvant or neoadjuvant systemic therapies to transplantation.

## 1. Introduction

Liver neoplasms are the sixth most common tumour worldwide and the third leading cause of cancer death. Hepatocellular carcinoma (HCC) makes up the majority of primary liver tumours, with over 90% of cases linked to cirrhotic liver disease. The main causes of chronic liver disease are hepatitis B and C viruses (HBV and HCV), alcohol, and increasingly, non-alcoholic steatohepatitis (NASH). Each year, 2–3% of patients with cirrhosis develop HCC [1].

The management of HCC is based on the Barcelona classification (BCLC) algorithm, which recommends interventions based on the tumour stage, liver function, and the patient’s overall condition [2]. Currently, HCC on a cirrhotic liver represents 30–35% of patient cases on the transplant waiting list, and it has become the second indication for liver transplantation in Spain, according to the National Transplant Organisation (ONT) [3].

Initially, liver transplantation for HCC in cirrhotic patients was based on Milan’s criteria (one nodule of ≤5 cm or up to three nodules of ≤3 cm), which resulted in a tumour recurrence risk of around 15%. In their classic 1996 study, Mazzaferro V et al. showed that endovascular therapy before transplantation offered no benefit if the Milan criteria were met. Additionally, the prognosis was better if the explant characteristics remained within these boundaries [4].

Advancements in surgical techniques, imaging, bridging therapies, and immunosuppression knowledge have led to successful HCC transplantation in patients meeting the Milan criteria. In 2001, the San Francisco criteria (UCSF) were published, offering similar survival and recurrence rates to those using Milan [5]. In 2008, Toso et al. [6,7] proposed using the total tumour volume (TTV) as a criterion for HCC transplantation. Building on this suggestion, Mazzaferro et al. in 2009 proposed the “up to seven” criteria, which achieve similar survival and recurrence rates as the original Milan criteria when the total tumour volume is less than seven centimetres [8].

More recently, in 2018, Mazzaferro et al. published Metroticket 2.0, which estimates individual survival and recurrence risks based on tumour size, the number of nodules and AFP levels [9,10]. Subsequently, other models, like MORAL, have since been developed that incorporate biological markers alongside radiological criteria. A pre-transplant MORAL considers AFP, the neutrophil-to-lymphocyte ratio (NLR) and maximum tumour size, while the post-transplant MORAL includes pathological data such as microvascular invasion, degree of tumour differentiation and actual tumour size [11]. These criteria have recently been validated by different groups [12].

Despite the new criteria, there are still some patients without macroscopic vascular involvement or distant tumours who are not candidates for liver transplantation at diagnosis. However, after systemic or percutaneous treatment, these patients can achieve radiological stability and an adequate tumour burden, making them eligible for liver transplantation (downstaging) [13,14]. The response to these treatments is standardized using RECIST 2.0 criteria. [15]. Since tumour behaviour over time reflects its biology, using locoregional therapy, followed by a waiting period before transplantation, can help predict this biology. The “ablate and wait” strategy involves performing locoregional therapy while the patient is on the transplant list and monitoring their response to assess tumour aggressiveness. Based on this approach, Kim et al. developed a predictive model for post-transplant recurrence risk. In this model, a score above 8 at diagnosis would exclude a patient from the list, and any increase in score during locoregional therapy would also result in exclusion from the list [16].

Among the factors related to HCC recurrence, alpha-fetoprotein (AFP) plays a significant role. AFP is a protein produced by hepatocytes during cell division. Numerous studies recommend actions based on AFP levels, such as contraindicating transplantation if AFP exceeds 1000 ng/mL and predicting a better prognosis if AFP is below 200 ng/mL at the time of transplantation [17,18,19]. In 2012, an AFP-based model was proposed, which classified patients according to their pre-transplant levels into <100, 100–1000 and >1000 ng/mL. This information, together with the number of nodules and the size of the largest nodule, allows for the estimation of survival and the risk of tumour recurrence [20].

Regarding factors observed in the removed liver tissue, microvascular invasion, capsule contact and tumour differentiation have been linked to the tumour’s biological behaviour [21]. 

According to the BCLC, locally advanced HCC may benefit from endovascular therapy (such as ablation, chemoembolization or radioembolisation) and patients could subsequently become eligible for liver transplantation [22,23,24,25]. 

Endovascular therapy is also recommended in patients meeting Milan’s criteria for whom the waiting time on the transplant list is longer than six months, as there is a risk of tumour progression [26].

Until recently, sorafenib and lenvatinib were the only systemic treatments available for HCC. However, treatments like atezolizumab-bevacizumab and durvalumab-tremelimumab are now widely used and have transformed the landscape of systemic HCC treatment [27,28,29,30].

Recent studies have explored new variables for predicting HCC recurrence. A nationwide study in China in 2018 identified cold ischaemia time as an independent factor for donor-related tumour recurrence [31]. Regarding the impact of HCV virus serostatus on HCC recurrence, recent studies found no differences between recipients with negative and positive serology, although this evidence is from a small cohort [32].

Donor age is another factor discussed in the literature, with some studies indicating that well-selected donors experience no increased risk of recurrence based on age [33].

Recently, new surrogate markers of tumour biological behaviour have been developed, such as: (1) the abnormal form of prothrombin induced by the absence of vitamin K (PIVKA-II, also known as DCP), which is produced during the malignant transformation of hepatocytes, and (2) the NLR [34,35]. Current research is exploring [18F] FDG/[18F] fluorocholine PET-CT as a marker of aggressiveness [36] and liquid biopsy as a predictor of micrometastases.

In relation to the above, the management of patients with HCC on liver transplant waiting lists, on which there are also patients with liver failure and non-tumour indications, is important to minimise the risk of tumour progression [37,38].

The purpose of this study is to share the insights gained from our transplant programme regarding liver transplantation for HCC and its recurrence, along with the factors that may be associated with it. Additionally, with over twenty years of data from liver transplant programmes, despite variations in criteria, indications, and techniques during this time, we find it pertinent to examine potential differences in the survival and recurrence rates of HCC between the two decades.

## 2. Materials and Methods

A single-centre retrospective cohort study was conducted. It included patients aged 18 and above who underwent liver transplantation for HCC between December 1997 and July 2020, which is the period of the transplant programme at our centre. Patient progress was followed up for at least one year. Those with incidental findings of HCC in the explant, liver tumours of histological types other than HCC and those lost to follow-up were excluded. Survival data were collected until 1 January 2023.

Demographic information, including patient sex, age at transplantation, duration on the waiting list, donor age, type of donation, and cirrhosis aetiology, was anonymously extracted from medical records and the liver transplant registry.

**In the first phase of the study**, we analysed the survival rates of patients who underwent liver transplantation for HCC and the overall rate of HCC recurrence. We also distinguished early recurrences that occurred within the first two years post-transplant.

Various factors related to HCC recurrence were examined:(1)Recipient-related factors: AFP levels at listing, the number of nodules, previous endovascular therapy, the size of the largest nodule, and the immunosuppression type used post-transplant (cyclosporine, tacrolimus, mycophenolate or everolimus/sirolimus).(2)Donor-related factors: cold ischaemia time, the type of donation (brain death/DCD (donor after cardio-circulatory death)), and donor age.(3)Characteristics of the explanted liver: capsule contact, microvascular invasion, and satellitosis.

To determine the AFP cut-off point for inclusion in the model, we constructed an ROC curve using AFP levels from the study sample. The cut-off point chosen for this study was 16 ng/mL AFP, as this provided a sensitivity of 71% and a specificity of 70%.

**The second phase of the study** involved comparing the survival and recurrence rates of HCC between two distinct periods within our transplant programme: 1997–2010 and 2011–2020.

Finally, we conducted an analysis of the percentage of early recurrence, both overall and between the two decades.

For the statistical analysis, we utilised the statistical software R 4.1.3 (version 4.1.3 for Windows^®^). The database variables were categorised into continuous quantitative variables (such as ischaemia time, recipient age, AFP, and lesion size) and categorical quantitative variables (including the number of lesions, the type of immunosuppression, and AP explant characteristics). The normality of continuous variables was assessed using the Shapiro–Wilk test, which revealed that none of them followed a normal distribution. Consequently, we employed the non-parametric Wilcoxon rank sum test to ascertain the differences between groups and calculate the relevant *p*-values. A significance level of *p* < 0.05 was adopted for all tests. The identified variables were utilised to construct a generalised logistic regression model (GLM) using the glm function from the stats package, as well as ROC curves with the pROC library. Additionally, we tested the homogeneity of demographic variables between the two groups (age, sex, aetiology of cirrhosis, and MELD) for homogeneity of variances, using the F-test. Furthermore, we assessed the survival of patients in both groups using the survminer and survival libraries and represented the results in a Kaplan–Meier table.

## 3. Results

### 3.1. Results of the First Phase of the Study

Of the 1094 liver transplants performed in our centre up to the closing date of the study, 265 were performed by HCC, which represents 24.22% of the total number of transplants performed. The demographic characteristics of the sample are summarized in Table 1.

The overall survival rate of patients was 68% at 5 years, 58% at 10 years, 45% at 15 years and 34% at 20 years. 

The overall HCC recurrence rate was 14.5%. Of these patients, 47% had recurrence at the extrahepatic level, 20.5% at the intrahepatic level, and 32.5% at both locations. Fifty-four per cent of the recurrences occurred early (before two years of follow-up). An analysis of the influence of the different variables on HCC recurrence is shown in Table 2a,b. Table 3 shows a breakdown of the types of bridging treatments used for the patients in the sample.

With those values that had reached statistical significance or showed a trend towards statistical significance, multivariate logistic regression analysis was performed, wherein a sensitivity of 85.7% and a specificity of 35.7% were found to predict recurrences in the samples analysed. The ROC curve generated from the model is shown in Figure 1.

### 3.2. Results of the Second Phase of the Study: Comparative Analysis between the Two Periods of the Transplant Programme

In total, 102 HCC transplants were performed in the first period and 157 in the second period. First, homogeneity analysis was performed to determine whether the two groups were comparable with respect to their baseline characteristics. All of them were homogeneous except for the aetiology of chronic liver disease; fewer CHC over HCV hepatopathy patients were detected in the second period compared to the first one. 

The survival rates of the patients were then compared between the first and second periods. No significant differences were found, but there was a trend towards longer long-term survival in patients transplanted in the second period (Figure 2).

The recurrence rate of HCC was 13.4% in the first period and 15.3% in the second (*p* = 0.6). The results of the analysis of the different variables in each of the periods are shown in Table 4a,b.

Finally, we found that 86% of early recurrences were detected in the second period. We determined that 50% of the early recurrences in the first period had poor prognostic pathology and that 53% of the early recurrences in the second period had unfavourable pathology in the explant, with no differences.

## 4. Discussion

This study aims to share insights from our transplant group. Similar studies have recently been published, aiming to analyse the factors linked to HCC recurrence [39]. In our series, 24.2% of the 1094 transplants performed were indicated for HCC, which is close to the 28.4% reported by the Spanish liver transplant registry (RETH) as of 2021 [40,41]. In the European transplant registry (ELITA), as of 2022, 17.58% of transplants were for HCC, probably due to the different degrees of development of the transplant programme in the different European countries, with transplantation due to liver failure being even more frequent [42].

Survival rates at 5 and 10 years in our cohort are slightly lower than those in the national transplant registry, but they equalize at 20 years. This trend might be attributed to higher mortality among transplant recipients in the early registry years, before refinements in surgical techniques and immunosuppression management.

Although European liver transplant registry data indicate increased survival for HCC indications across the transplantation periods, our study did not observe such differences. This may be because the registry encompasses diverse countries with distinct strategies and levels of transplant programme development. Our data align more closely with those of the Spanish registry [40,41].

Regarding HCC recurrence in our series, it mirrors that of other large published series and meets the quality criteria set by the Milan criteria and subsequent criteria [43,44]. Our study developed a model predicting HCC recurrence in transplant recipients based on AFP levels, nodule count, nodule size and explant characteristics, achieving 85.7% sensitivity and 35.7% specificity.

We reinforced the strategies employed in our transplant programme over time, maintaining our results by expanding our inclusion criteria and enhancing flexibility in terms of donor and recipient age. For analysing factors like the influence of cold ischaemia time on HCC recurrence, we used a six-hour cut-off point based on the literature and surgical guidelines [45]. 

Although it was not significant in our study, several papers have recently been published arguing that prolonged cold ischaemia time is an independent factor for HCC recurrence, and that hypothermic oxygen perfusion in-do-ischaemic oxygen perfusion (HOPE) devices may decrease the risk of HCC recurrence [46,47].

To determine the influence of donor age, we chose a cut-off point of 65 years, similar to other studies [48]. Regarding the benefit of using organs from donors older than 65 years, the results have been similar to previous studies, as there is no increased risk of recurrence and they provide a similar survival benefit for patients as long as the organs are carefully selected [49,50]. In our cohort, no lower graft survival was identified when the recipients were older than 65 years, although there are studies with mixed results on this aspect, such as that of Lerosey et al., who, in a large follow-up series, found a lower 5-year survival rate in transplanted patients older than 65 years compared to younger patients [51], while other groups such as Khalayleh et al. or Gomez-Navarra et al. presented large series in which transplanted patients who were over 65 years of age had similar survival rates [52,53].

The analysis of the anatomopathological characteristics of the explant was divided into low-risk and high-risk. High risk was defined as those explants in which microvascular or lymphatic invasion, poorly differentiated carcinoma, satellitosis and the contact of an HCC nodule with the capsule were observed, as these factors had been identified as independent factors for HCC recurrence, especially in hepatectomies [54].

In our cohort, we have no patients who were treated with TARE therapy prior to transplantation. No differences were found regarding HCC recurrence with respect to having undergone pre-transplant therapy or not. Recently, there have been significant developments and research into downstaging therapies. There are already some studies that combine systemic therapy with percutaneous therapy, allowing patients to be rescued for surgery and offering similar survival rates to patients with less advanced stages at diagnosis [28].

With respect to AFP, the cut-off point for normality in the clinical analysis laboratory of our centre is 8 ng/mL, so, based on the results of the ROC curve generated from the sample data, it was decided to use the point of 16 ng/mL as the clinically relevant elevation point, performing the analysis on this basis and obtaining a sensitivity of 75% and a specificity of 69%. Previous studies have proposed a cut-off of 10 ng/mL, giving a sensitivity of 66.3% and a specificity of 80.6% [55]. Other studies have proposed 20 ng/dL with a sensitivity of 41–65% and a specificity of 80–94% [56].

Although there are recent studies on the survival benefit of HCC transplant recipients treated with everolimus [57], the statistically significant differences identified in our study with respect to the type of immunosuppression are difficult to interpret and are probably associated with the small sample size for less commonly used first-line drugs such as mycophenolate and everolimus, compared to cyclosporine and tacrolimus.

Recent studies have related MELD at the time of transplantation to transplantation with the risk of HCC recurrence, which could be an interesting parameter to analyse in future studies and which has not been performed in this study, due to the significant lack of data for patients in the first period [58].

As in our study, there are precedents where the application of the up-to-seven criterion, especially in combination with AFP levels, shows similar recurrence rates to those obtained when the Milan criteria alone were applied [44,59].

As a novelty, our study presents a comparison between extended periods (1996–2010 and 2010–2020). There are no precedents for transplant groups where the results have been evaluated in terms of survival and recurrence compared to previous periods. No significant differences were found between the two groups in terms of survival or the recurrence of HCC, although there was a trend towards the increased survival of transplant patients in the second period and a paradigm shift with respect to the aetiology of liver disease and serostatus at the time of transplantation, thanks to the generalisation of HCV treatments. The failure to find a significant reduction in the percentage of recurrences, as expected, is probably related to the fact that recurrences of between 8 and 15% are expected within the transplant criteria used, these values not being sufficiently wide for there to be significant differences.

In the comparison, a higher percentage of early relapses in the second period of the study is probably related to the active search and close follow-up of patients with a higher risk of relapse, with detection earlier than in the previous period. 

Currently, the optimisation of viral disease management [60] and the standardisation of screening programmes, in part due to increased public awareness, are allowing the early detection of HCC and offering early-stage interventions to the patient, resulting in a greater survival benefit. 

Despite the significant progress in HCC management in recent years, early detection strategies and therapeutics need to be improved to achieve better survival rates. The improvement of imaging techniques and research into new tumour markers are some of the fields undergoing major development. Conversely, recurrence after resection, ablation or transplantation requires new studies on adjuvant therapies [61].

Perhaps in the future, thanks to the new systemic agents available, it will be an option to consider adjuvant systemic treatment in patients with poor prognostic factors and a high risk of tumour recurrence. 

Despite all these advances, we cannot forget the challenge of managing liver transplant lists. Recently, some strategies based on artificial intelligence have been developed for organ prioritisation, which, together with the new and increasingly accurate relapse prediction models, offer very promising perspectives for the organisation of transplant programmes [62,63].

The main limitation of this study is its single-centre, observational and retrospective nature. The long follow-up period is a strength of the present work but is also a limitation, since there may be difficulties in retrieving data from older records. For the same reason, basing research on clinical practice data is both a strength and a limitation. On the other hand, our centre is a reference centre for liver transplantation, which has allowed us to recruit a larger number of patients than in other published studies. For all these reasons, we believe that our experience can contribute to the improvement of knowledge on the subject and to the improvement of treatments for these patients.

## 5. Conclusions

The application of the criteria established in the clinical practice guidelines for the selection of candidates for liver transplantation and their prioritisation on waiting lists allows, in real clinical practice, the obtaining of results that are adjusted to quality standards and are similar in different working groups at both national and international levels.

According to our results, patients transplanted within certain criteria (Milan or up-to-seven) maintain a recurrence rate and survival benefit within the expected survival benefit based on previous studies.

The early recurrence rate justifies a close surveillance strategy in the first two years post-transplantation.

Alpha-fetoprotein levels at transplantation, the total tumour size and the number of tumour nodules allow individualisation of the risk of recurrence, offering these patients stricter follow-up strategies and the early adjustment of immunosuppression.

Given the important developments in this field, it is likely that, in the near future, the detection of patients at high risk of recurrence will allow the indication of adjuvant or neoadjuvant systemic therapies in transplantation.

Similar studies with large series and the comparison of results throughout the transplantation programme would provide interesting information to assess the strategies implemented over the years.

## Figures and Tables

**Figure 1 biomedicines-12-01302-f001:**
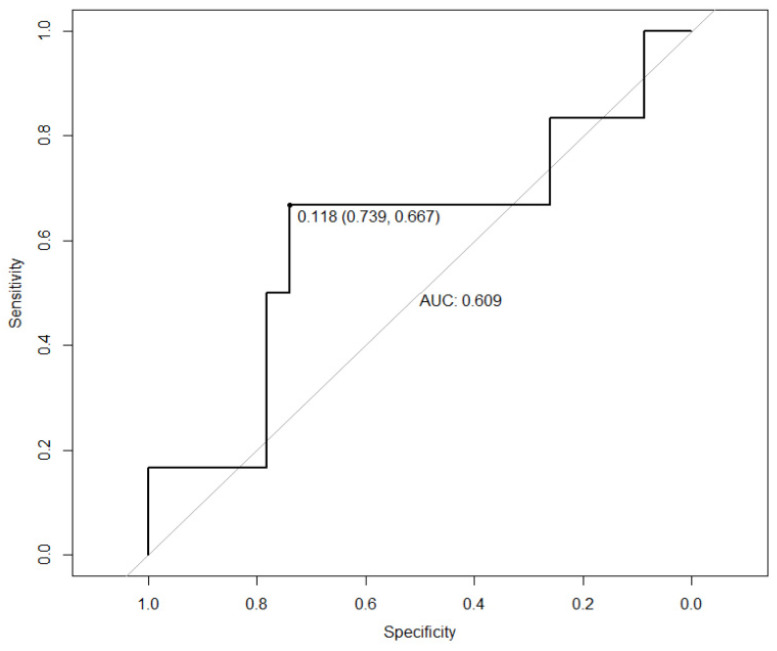
ROC curve constructed from the data obtained from a logistic regression performed with those variables that have reached statistical significance.

**Figure 2 biomedicines-12-01302-f002:**
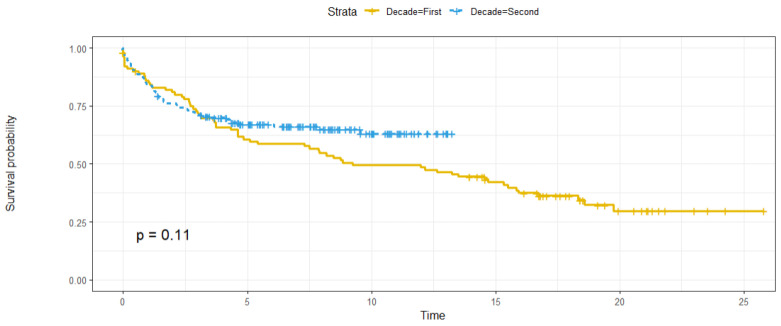
Kaplan–Meier plot comparing the survival rates of transplanted patients in both periods. First period: 1997–2010. Second period: 2010–2020.

**Table 1 biomedicines-12-01302-t001:** Demographic characteristics of the sample.

Variable	*N* = 265
Sex of recipients Male	79%
Female	21%
Average age at Transplantation (years)	58 ± 7.5
Cirrhosis aetiologies HCV	42%
Alcohol	23%
HBV	11%
Virus D co-infection	1.5%
HCV + alcohol	13.2%
HBV + alcohol	1.5%
HCV + VHB	1.5%
NASH	0.75%
Cryptogenetics	3%
CBP	0.35%
Autoimmune hepatitis	1.5%
Haemochromatosis	0.7%
Average age of donors (years)	58.7 ± 14.99
Type of donation	
Brain death	94%
Asystole	6%
MELD at transplantation	11.8 ± 4.95
Criteria	
Milan	89.27%
Up-to-seven	9.6%
Downstaging	1.13%
Previous endovascular treatment	85.7%
Immunosuppression	
Cyclosporine	23%
Tacrolimus	59.1%
Mycophenolate	5.3%
Everolimus/sirolimus	5.5%
Tacrolimus + everolimus	7.1%

HCV: hepatitis C virus; HBV: hepatitis B virus; NASH: non-alcoholic steatohepatitis; CBP: primary biliary cholangitis. This table describes the demographic characteristics of the whole sample being analysed.

**Table 2 biomedicines-12-01302-t002:** (a) Analysis of continuous variables for post-transplant HCC recurrence. (b) Analysis of categorical variables with respect to post-transplant HCC recurrence.

**(a)**					
**Variable**	**Recurrence Group Values**	**Non-Recurrence Group Values**		** *p-* ** **Value**	
Cold ischaemia time (minutes)	393.94 ± 235.96	349.43 ± 177.49		0.35	
Age of recipient	57.69 ± 7.42	58.10 ± 7.44		0.84	
Age of donor	57.8 ± 14.45	59.02 ± 14.76		0.74	
Pre-transplant alpha protein > 16 ng/mL	229.85 ± 363.05	40.74 ± 117.15		0.00085 **	
Larger nodule	31.96 ± 12.97	27.31 ± 13.28		0.062 *	
Cold ischaemia time	393.94 ± 235.96	349.43 ± 177.49		0.35	
**(b)**					
**Variable**		**Recurrence** **Group Values**	**Non-Recurrence** **Group Values**		***p*** **Value**
Number of lesions	1 >1	44.4 55.6	65.7 34.3		0.06958
AP explant characteristics	Good prognosis Poor prognosis	56 44	84 16		0.003905 **
Immunosuppression	Cyclosporine Tacrolimus Mycophenolate Everolimus/sirolimus Tacrolimus + everolimus	40.5 25.9 0.3 14.8 18.5	59.1 15.9 11.4 5.3 8.3		0.02729 **
Previous endovascular treatment		87	78		0.53

* Trend towards statistical significance. ** Statistical significance.

**Table 3 biomedicines-12-01302-t003:** Description of the treatments performed prior to transplantation as bridging therapy.

	*N* = 227
TACE	63%
ARF	25.2%
Hepatectomy	3.1%
ARF + TACE	6.1%
Hepatectomy + TACE	1.8%
Hepatectomy + ARF	0.4%
External radiation therapy	0.4%

TACE: trans-arterial chemoembolization; ARF: radiofrequency ablation.

**Table 4 biomedicines-12-01302-t004:** (a) Comparative analysis of the different variables in each of the periods: continuous variables. (b) Comparative analysis of the different variables in each of the periods: categorical variables.

**(a)**			
**Variable**	**1st Period**	**2nd Period**	***p*** **Value**
Larger nodule	26 ± 13.5	28.67 ± 13.22	0.1889
Pre-transplant alpha protein > 16 ng/ml	41.96 ± 72.99	77.52 ± 15.18	0.077
Age of recipient	56.52 ± 7.789	58.99 ± 6.98	0.021
Donor age **	-	-	-
Cold ischaemia time **	*-*	*-*	*-*
**(b)**			
**Variable**	**1st Period**	**2nd Period**	***p*** **Value**
Number of lesions			
1	64.6	58.7	0.088
>1	35.4	41.3	
AP explant characteristics			
Good prognosis	73.7	81.3	
Poor prognosis	26.3	18.7	0.44
Immunosuppression			0.00007
Cyclosporine	19.5	17.2	
Tacrolimus	44.4	59.3	
Mycophenolate	33.3	2.4	
Everolimus/sirolimus	2.8	8.1	
Tacrolimus + everolimus	0	13	

** Not analysed due to a lack of sufficient data in the first period.

## Data Availability

Data are contained within the article.

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
