# Peer review of "Liver Transplantation for Hepatocarcinoma: Results over Two Decades of a Transplantation Programme and Analysis of Factors Associated with Recurrence"

_biomedicines, 2024, doi:10.3390/biomedicines12061302_

Round 1

Reviewer 1 Report

Comments and Suggestions for Authors

1. line 10: e-mail address of the corresponding author is incomplete

2. lines 17-18: there is a repetition

3. Table 1: abbreviations used should be explained below the table; change "VHC" into "HCV;" change "VHB" into "HBV;" correct typos: autoinmune, haemocrhomatosis

4. Table 2a and Figure 1: I would avoid the term "trend towards (or tendency to) statistical significance"

5. Table 2b: p value is lacking

6. Table 3: abbreviations used should be explained below the table

7. carefully check the References, e.g. 37, 38

Comments on the Quality of English Language

Language edition performed by an English native speaker should be done.

Reviewer 2 Report

Comments and Suggestions for Authors

The current article “Liver transplantation for hepatocarcinoma: results over two decades of a transplantation programme and analysis of factors associated with recurrence” by Martínez Burgos et al present the long-term analysis of the survivor to determine the effect of different factors in recurrence of hepatocarcinoma in context to liver transplant program.

I have few comments of the current form of manuscript to improve the overall quality of article:

Line 96- Thanks to this…… Authors should rewrite the whole statement

Line 99- 102 Based on this good response…. The statement is confusing and does not make any sense. Authors should rewrite the whole statement

Line 116- impact of C virus… Typo or Authors meant to write HCV?

Table 2A- Cold ischaemia time? What is the difference in top and bottom annotation?

Table 2b have no significant analysis while in table 4b, there is p-value? Any specific reason for this inconsistency?

Table 3- Total percentage is 100.1 not 100%. Whether the data have any addition or over-expectation.

There are many limitations in the study and result section. Authors have not provided the comprehensive analysis on this important clinical research topic. Results were inconsistence and not performed in curated way. Authors seems in urgency to publish article without providing proper clinical information and methodology for the study.

Author Response

"Please see the attachment

Round 2

Reviewer 2 Report

Comments and Suggestions for Authors

The authors addressed all my concerns. I understand the limitation of study. Therefore, I would like to ask authors to include limitations also in manuscript.
